# Heritable Connective Tissue Disorders in Childhood: Increased Fatigue, Pain, Disability and Decreased General Health

**DOI:** 10.3390/genes12060831

**Published:** 2021-05-28

**Authors:** Jessica Warnink-Kavelaars, Lisanne E. de Koning, Lies Rombaut, Mattijs W. Alsem, Leonie A. Menke, Jaap Oosterlaan, Annemieke I. Buizer, Raoul H. H. Engelbert

**Affiliations:** 1Department of Rehabilitation Medicine, Amsterdam Movement Sciences, Amsterdam UMC, University of Amsterdam, Meibergdreef 9, 1105 AZ Amsterdam, The Netherlands; m.w.alsem@amsterdamumc.nl (M.W.A.); ai.buizer@amsterdamumc.nl (A.I.B.); r.h.engelbert@amsterdamumc.nl (R.H.H.E.); 2Center of Expertise Urban Vitality, Faculty of Health, Amsterdam University of Applied Sciences, Tafelbergweg 51, 1105 BD Amsterdam, The Netherlands; l.e.de.koning@hva.nl; 3Center of Medical Genetics, Ghent University Hospital, Corneel Heymanslaan 10, 9000 Ghent, Belgium; lies.rombaut@ugent.be; 4Department of Pediatrics, Emma Children’s Hospital, University of Amsterdam, Amsterdam UMC, Meibergdreef 9, 1105 AZ Amsterdam, The Netherlands; l.a.menke@amsterdamumc.nl; 5Emma Neuroscience Group, Amsterdam Reproduction & Development, Department of Pediatrics, Emma Children’s Hospital, University of Amsterdam, Amsterdam UMC, Meibergdreef 9, 1105 AZ Amsterdam, The Netherlands; j.oosterlaan@amsterdamumc.nl; 6Department of Rehabilitation Medicine, Rehabilitation and Development, Amsterdam Movement Sciences Institute, Amsterdam UMC, Vrije University of Amsterdam, De Boelelaan 1117, 1081 HV Amsterdam, The Netherlands; 7Department of Clinical Genetics, Amsterdam UMC, University of Amsterdam, Meibergdreef 9, 1105 AZ Amsterdam, The Netherlands; 8Department of Genetics, University Medical Center Groningen, Hanzeplein 1, 9713 GZ Groningen, The Netherlands; 9Center for Human and Clinical Genetics, Leiden University Medical Center, Albinusdreef 2, 2333 ZA Leiden, The Netherlands; 10Department of Human Genetics, Radboud University Medical Center, Geert Grooteplein Zuid 10, 6525 GA Nijmegen, The Netherlands; 11Department of Clinical Genetics, Maastricht University Medical Center, P. Debyelaan 25, 6229 HX Maastricht, The Netherlands; 12Centre of Medical Genetics, Faculty of Medicine and Health Sciences, University of Antwerp and Antwerp University Hospital, Drie Eikenstraat 655, 2650 Edegem, Belgium; 13Department of Physical and Rehabilitation Medicine, Child Rehabilitation, Ghent University Hospital, Corneel Heymanslaan 10, 9000 Ghent, Belgium

**Keywords:** Heritable Connective Tissue Disorders, Marfan syndrome, Ehlers-Danlos syndromes, hypermobile Ehlers-Danlos syndrome, Loeys-Dietz syndrome, fatigue, pain, general health, disability, children

## Abstract

Heritable Connective Tissue Disorders (HCTD) show an overlap in the physical features that can evolve in childhood. It is unclear to what extent children with HCTD experience burden of disease. This study aims to quantify fatigue, pain, disability and general health with standardized validated questionnaires. Methods. This observational, multicenter study included 107 children, aged 4–18 years, with Marfan syndrome (MFS), 58%; Loeys-Dietz syndrome (LDS), 7%; Ehlers-Danlos syndromes (EDS), 8%; and hypermobile Ehlers-Danlos syndrome (hEDS), 27%. The assessments included PROMIS Fatigue Parent–Proxy and Pediatric self-report, pain and general health Visual-Analogue-Scales (VAS) and a Childhood Health Assessment Questionnaire (CHAQ). Results. Compared to normative data, the total HCTD-group showed significantly higher parent-rated fatigue T-scores (*M* = 53 (*SD* = 12), *p* = 0.004, *d* = 0.3), pain VAS scores (*M* = 2.8 (*SD* = 3.1), *p* < 0.001, *d* = 1.27), general health VAS scores (*M* = 2.5 (*SD* = 1.8), *p* < 0.001, *d* = 2.04) and CHAQ disability index scores (*M* = 0.9 (*SD* = 0.7), *p* < 0.001, *d* = 1.23). HCTD-subgroups showed similar results. The most adverse sequels were reported in children with hEDS, whereas the least were reported in those with MFS. Disability showed significant relationships with fatigue (*p* < 0.001, *r_s_* = 0.68), pain (*p* < 0.001, *r_s_* = 0.64) and general health (*p* < 0.001, *r_s_* = 0.59). Conclusions. Compared to normative data, children and adolescents with HCTD reported increased fatigue, pain, disability and decreased general health, with most differences translating into very large-sized effects. This new knowledge calls for systematic monitoring with standardized validated questionnaires, physical assessments and tailored interventions in clinical care.

## 1. Introduction

Heritable Connective Tissue Disorders (HCTD) are characterized by pathological connective tissue fragility in multiple organ systems. The diagnosis is based on clinical criteria and/or molecular confirmation by a causative genetic variant [1,2,3]. The phenotypes of the most common HCTD, Marfan syndrome (MFS) [1], Loeys-Dietz syndrome (LDS) [2] and Ehlers-Danlos syndromes [3], show an overlap in the musculoskeletal, cardiovascular and cutaneous features [1,2,3,4,5,6,7] that can evolve in childhood. It is unclear to what extent children and adolescents with HCTD experience physical impairments, limitations in activities and burden of disease, and whether there is a difference among HCTD-subgroups [8].

In previous qualitative semi-structured interview studies, parents of children with MFS (4–12 years) and adolescents with MFS (12–18 years) experienced problems of physical functioning, participation in activities and daily life and keeping up with peers [9,10]. Furthermore, studies in children with MFS and hEDS have reported fatigue and pain to negatively impact daily (physical) functioning [7,11,12,13,14,15,16,17,18,19], a high incidence of pain-related disability [17] and deteriorating physical functioning over time [20]. To our best knowledge, no quantitative studies, using validated questionnaires, have been conducted into pain, fatigue and disability in children with MFS, LDS and molecularly confirmed types of EDS (hereafter EDS). A few studies in children and adolescents with hEDS and Hypermobility Spectrum Disorders (HSD, the current label for patients with joint hypermobility and musculoskeletal complications who do not fulfil the criteria for hEDS) [7] reported on increased fatigue and pain [18], generalized hyperalgesia [21], and improvement of disability after following an outpatient multidisciplinary rehabilitation treatment program [17]. Regarding adults with HCTD, it has been reported that patients with MFS, LDS, EDS and hEDS have persistent fatigue, pain, disability and burden of disease [4,5,6,14,16,22,23,24,25,26,27,28]. 

This study aims to gain better insight into the prevalence and severity of fatigue, pain, disability and general health in children and adolescents diagnosed with the most common HCTD using standardized validated questionnaires. 

## 2. Methods

### 2.1. Study Design

This study was an observational cross-sectional multicenter survey study.

### 2.2. Participants 

Those eligible for inclusion were all children and adolescents, aged 4–18 years, with MFS [1], LDS [2], EDS [3], hEDS [3] and HSD [7]. 

### 2.3. Procedures 

The Expert Centers for Marfan syndrome and related Connective Tissue Disorders in the Netherlands, including the University Medical Centers of Amsterdam, Leiden, Groningen, Maastricht and Nijmegen, and the Center for Medical Genetics of the Ghent University Hospital in Belgium, included participants in the study. Additionally, the MFS and EDS patient associations in the Netherlands announced the study information on their websites. Participants could contact the research team by email or phone. Children and parents were invited by letter and written informed consent was obtained. The study took place from February 2019 to March 2020. The survey completion time was 80–95 min. The Medical Ethics Review Committee of the Amsterdam UMC (reference number W18_346) and the Ethical Committee of Ghent University Hospital (EC2019/1958) approved the study’s protocol.

### 2.4. Participant Characteristics

A custom-made parent questionnaire collected information on the sex, age, nationality and HCTD diagnosis of their child and on their own sex, age and nationality. One of the parents completed the questionnaire.

### 2.5. PROMIS Fatigue Pediatric Self-Report and Parent Proxy

The Patient Reported Outcomes Measurement Information System’s (PROMIS) Fatigue 10a Pediatric v2.0 short form and Fatigue 10a Parent Proxy v2.0 short form assess self-reported fatigue in children aged 8-18 years, and parent reported fatigue in children under eight, respectively. Both questionnaires contain 10 fatigue statements that pertain to the degree of fatigue and the impact of fatigue on physical, mental and social activities, as experienced during the last seven days. Each question has five response options (never = 1, rarely = 2, sometimes = 3, often = 4, always = 5). To calculate the total raw score, the values of the response to each question are summed and then rescaled into a standardized T-score with a mean of 50 and a standard deviation (SD) of 10. Both questionnaires are widely used rating scales with well-studied and excellent psychometric properties, and they discriminate well between disease severity [29,30,31]. 

### 2.6. Childhood Health Assessment Questionnaire (CHAQ), Pain VAS and General Health VAS

The Dutch version of the CHAQ [32] assesses functional ability in daily life activities and distinguishes between the following eight domains: dressing, arising, eating, walking, hygiene, reach, grip and activities (30 items). The response scores for each item range from 0–3. The highest score of an item within a domain determines the domain score. The utilization of assistance or aids in a domain sets the domain score to a minimum of two. The mean score of the eight domains determines the CHAQ disability index (CHAQ-DI) and ranges from zero (no disability) to three (disabled) [32,33]. Children under eight are proxy-reported. Children aged 8–18 years self-report. The pain and general health Visual Analogue Scales (VASs) supplement the CHAQ. The pain VAS assesses subjective pain over the last week. The intensity of pain is scored on a 0–100 scale, with zero referring to “no pain” and 100 to “very severe pain”. Children under eight are proxy-reported. Children aged 8-18 years self-report. The general health VAS assesses current subjective general health. General health is scored on a 0–100 scale, with zero referring to “very good general health” and 100 to “very poor general health”. Children aged 4–18 years are proxy reported. The CHAQ, pain VAS and general health VAS are widely used rating scales with well-studied and excellent psychometric properties, and they discriminate well between children with chronic conditions and healthy children [32,33,34,35]. 

### 2.7. Statistical Analysis 

Online survey data were exported from the Castor database to the Statistical Package for Social Science (SPSS) version 26.0. Data were analyzed as the total HCTD-group and separately, as the HCTD-subgroups: MFS, LDS, EDS and hEDS. The group sizes of the HCTD-subgroups EDS (n = 9) and LDS (n = 7) were small, and the analyses were for explorative interpretation only. 

Data were checked for errors, missings and outliers. Sex, age and nationality of the HCTD-group, HCTD-subgroups and of the parent, who completed the survey, were analyzed using descriptive analyses. To compare the normative categorical data, age-groups, sex and nationality, of the PROMIS Fatigue, CHAQ, pain VAS and general health VAS [32,33] to the HCTD-group and HCTD-subgroups data, chi–square tests were used and presented as Odds Ratios (OR) and 95% confidence intervals (CI) [36]. 

The normality of the distributions was visually inspected using normality plots and tested using Shapiro–Wilk tests. The CHAQ-DI scores, pain VAS scores and general health VAS scores of the HCTD-group and HCTD-subgroups were not all distributed normally; these data were reported using the median and interquartile range (IQR). However, normative data of these questionnaires have been reported using means and standard deviation (SD). Consequently, for comparison reasons, our data were also reported by means (SD). To compare the normative scores of the PROMIS Fatigue, CHAQ-DI, pain VAS and general health VAS [31,33] to the HCTD-group’s and HCTD-subgroup’s scores, independent-sample *t*-tests were used. Among the HCTD-subgroups, Kruskal–Wallis tests and Mann–Whitney U tests were used. 

Severe fatigue was defined as a standardized T-score > 70, based on the PROMIS Fatigue normative T-scores [31,36,37]. Severe disability was defined as a standardized Z-score of <−2, based on the normative CHAQ-DI scores [32,33,36]. To compare the severe fatigue and severe disability percentages of the normative scores to the HCTD-group and HCTD-subgroups, chi–square tests were used. The effect sizes were calculated. For parametric tests, Cohen’s *d* was defined as the difference between the mean and the normative data mean, divided by the pooled standard deviations, with values of 0.2, 0.5 and 0.8 defined as the thresholds for small, moderate and large effects, respectively [36]. For non-parametric tests, Mann–Whitney U test’s *r* was defined as the Z-score divided by the square root of observations, with values of 0.1, 0.3 and 0.5 defined as the thresholds for small, moderate and large effects, respectively. Spearman’s rho *r_s_*, a non-parametric test, was used to explore relationships between fatigue, pain, disability and general health in the HCTD-group and the HCTD-subgroups, where the value *r_s_* = 1 means a perfect positive correlation and the value *r_s_* = −1 means a perfect negative correlation [36]. A *p*-value ≤ 0.05 was considered statistically significant [36]. 

## 3. Results 

### 3.1. Participants

Table 1 shows the sex, age and nationality of children of the HCTD-group; the HCTD-subgroups: MFS, LDS, EDS and hEDS; and of the parents who completed the survey. Initially, 156 children and adolescents agreed to participate, of whom five were not diagnosed with HCTD. They were excluded. Another 44 participants did not complete the survey and, consequently, there are no data related to these participants. In total, 107 children and adolescents with HCTD participated: MFS, 58%; LDS, 7%; EDS, 8%; hEDS, 27%; and HSD, 0%. The mean age (SD) was 10.0 (4.1) years and 55% of the children were male.

#### 3.1.1. Fatigue Pediatric self-Report 

Table 2 and Figure 1 show the PROMIS Fatigue 10a Pediatric v2.0 short form and the PROMIS Fatigue 10a Parent Proxy v2.0 short form T-scores of the normative data, the HCTD-group and the HCTD-subgroups, MFS, LDS, EDS and hEDS; comparisons between the normative T-scores and the HCTD-group’s T-scores; and comparisons among the HCTD-subgroup’s T-scores.

Compared to normative T-scores, the HCTD-group did not differ significantly, indicating no increased fatigue; the HCTD-subgroup hEDS reported significantly higher T-scores, translating into a large-sized effect, indicating increased fatigue (*p* < 0.001, *d* = 1.1); and the HCTD-subgroup MFS reported significantly lower T-scores, translating into a large-sized effect, indicating decreased fatigue (*p* < 0.001, *d* = 0.6). Compared to normative data, the percentage of children with a T-score above the severe fatigue cut-off was significantly greater in the HCTD-subgroups EDS and hEDS (*OR* 6.6, 95% *CI* 2.5–19.6, *p* = 0.03; *OR* 6.6, 95% *CI* 1.2–36.5, *p* < 0.001, respectively). 

Explorative analyses of fatigue among the HCTD-subgroups showed significantly higher fatigue scores for in the hEDS-subgroup compared to the MFS-subgroup and LDS-subgroup, translating into large-sized effects (*p* < 0.001, *r* = 0.61 *; p* = 0.006, *r* = 0.61, respectively).

#### 3.1.2. Fatigue Parent-Proxy 

Compared to normative T-scores, the HCTD-group reported significantly higher T-scores, translating into a small to medium-sized effect (*p* = 0.004, *d* = 0.3); and the HCTD-subgroup hEDS also reported significantly higher T-scores, translating into a very large-sized effect (*p* < 0.001, *d* = 1.4). Both results indicate increased fatigue. The other HCTD-subgroups showed no significant differences. Compared to normative data, the percentage of children with a T-score above the severe fatigue cut-off was significantly greater for the HCTD-group and HCTD-subgroups EDS and hEDS (*OR 2.8*, *95% CI 1.6–4.9*, *p* < 0.001; *OR* 6.7, 95% *CI* 1.7–27.0, *p* < 0.001; *OR* 10.5, 95% *CI* 4.7–23.5, *p* < 0.001, respectively). 

Explorative analyses of fatigue among the HCTD-subgroups showed significantly higher fatigue scores for the hEDS-subgroup compared to the MFS-subgroup and LDS-subgroup, translating into large-sized effects (*p* < 0.001, *r* = 0.57; *p* < 0.001, *r* = 0.51, respectively).

#### 3.1.3. Disability

Table 3 and Figure 1 show the Childhood Health Assessment Questionnaire (CHAQ), pain VAS and general health VAS: CHAQ domain scores, CHAQ disability index scores, pain VAS scores and general health VAS scores of the normative data, the HCTD-group and the HCTD-subgroups: MFS, LDS, EDS and hEDS; comparisons between the normative scores and the HCTD-group’s scores; and comparisons among the HCTD-subgroup scores.

Compared to normative CHAQ-DI scores, the HCTD-group and all of the HCTD-subgroups reported significantly higher scores, translating into medium to very large-sized effects [32], indicating increased disability (*HCTD* (*p* < 0.001, *d* = 1.23), *MFS* (*p* < 0.001, *d* = 0.78), *LDS* (*p* = 0.003, *d* = 1.10), *EDS* (*p* < 0.001, *d* = 1.11), *hEDS* (*p* < 0.001, *d* = 2.28))). Compared to normative data, the percentage of children with a score above the severe disability cut-off was significantly greater for the HCTD-group and all of the HCTD-subgroups (*HCTD* (*OR* 24.3, 95% *CI* 5.6–104.7, *p* < 0.001), *MFS* (*OR* 11.3, 95% *CI* 2.4–52.2, *p* < 0.001), *LDS* (*OR* 15.6, 95% *CI* 1.8–135.1, *p* = 0.001), *EDS* (*OR 31.2*, *95% CI 4.6–213.5*, *p* < 0.001), *hEDS* (*OR* 123.5, 95% *CI* 23.1–660.7, *p* < 0.001), respectively). Children with HCTD with severe disability (*Z*-score ≤ −2) did not differ significantly with respect to age (*p* = 0.43) or sex (*p* = 0.58) compared to children without disability (*Z*-score ≥ 0). 

Explorative analyses of disability (CHAQ-DI) among the HCTD-subgroups showed significantly higher disability scores, translating into medium to large-sized effects in the hEDS-subgroup compared to MFS-subgroup, LDS-subgroup and EDSsubgroup (*p* < 0.001, *r* = 0.58; *p* = 0.005, *r* = 0.46; *p* = 0.043, *r* = 0.32, respectively). 

The assistance of another person was required in 41% in the HCTD-group participants (MFS, 27%; LDS, 28%; EDS, 33 %; and hEDS, 76%), mainly for errands and chores. The utilization of aids or devices during activities was needed in 46% (MFS, 29%; LDS, 57%; EDS, 56%; and hEDS, 80%). Special utensils and a wheelchair were used the most.

#### 3.1.4. Pain

Compared to normative pain VAS scores, the HCTD-group and all of the HCTD-subgroups reported significantly higher scores, translating into medium to very large-sized effects [33], indicating a higher pain intensity during the last week (*HCTD* (*p* < 0.001, *d* = 1.27), *MFS* (*p* < 0.001, *d* = 0.80), *LDS* (*p* < 0.001, *d* = 1.46), *EDS* (*p* < 0.001, *d* = 2.30), *hEDS* (*p* < 0.001, *d* = 3.17)). 

Explorative analysis of the pain VAS scores among the HCTD-subgroups showed a significantly higher pain intensity translating to medium to large-sized effects for the hEDS subgroup compared to the MFS-subgroup, LDS-subgroup and EDS-subgroup (*p* < 0.001, *r* = 0.63; *p* = 0.016, *r* = 0.54; *p* = 0.021, *r* = 0.40, respectively). The group of children with HCTD with a reported pain VAS score (n = 93) compared to the group of children without a reported pain VAS score (missings = 14), did not differ significantly with respect to age (*p* = 0.98) or sex (*p* = 0.46).

#### 3.1.5. General Health

Compared to normative General health VAS scores, the HCTD-group and all of the HCTD-subgroups reported significantly higher scores, translating into very large-sized effects [32], indicating decreased general health (*HCTD* (*p* < 0.001, *d* = 2.04), *MFS* (*p* < 0.001, *d* = 2.1), *LDS* (*p* < 0.001, *d* = 2.0), *EDS* (*p* < 0.001, *d* = 1.9) and *hEDS* (*p* < 0.001, *d* = 3.3)). 

Explorative analysis of general health VAS scores among the HCTD-subgroups showed significantly decreased general health, translating into medium to large-sized effects, for the hEDS-subgroup compared to the MFS-subgroup and LDS-subgroup (*p* < 0.001, *r* = 0.58; *p* = 0.025, *r* = 0.42, respectively). The group of children with HCTD with a reported general health VAS score (n = 82) compared to the group of children without a reported general health VAS score (missings = 25), were not significantly different with respect to age (*p* = 0.140) or sex (*p* = 0.72).

#### 3.1.6. Correlations 

Disability in the HCTD-group was significantly positively correlated to fatigue, in both parent-proxy and pediatric self-report responses (*r_s_* = 0.65, *p* < 0.001; *r_s_* = 0.72, *p* < 0.001, respectively), pain (*r_s_* = 0.60, *p* < 0.001) and general health (*r_s_* = 0.58, *p* < 0.001). General health in the HCTD-group was significantly positively correlated to fatigue, in both parent-proxy and pediatric self-report (*r_s_* = 0.66, *p* < 0.001; *r_s_* = 0.7, *p* < 0.001, respectively) and pain (*r_s_* = 0.72, *p* < 0.001). Pain in the HCTD-group was significantly positively correlated with fatigue, both in parent-proxy and pediatric self-report responses (*r_s_* = 0.63, *p* < 0.001, *r_s_* = 0.68, *p* < 0.001, respectively).

Explorative analysis in the HCTD-subgroup MFS showed that disability was moderately significantly positively correlated to fatigue reported by parent-proxy and general health (*r_s_* = 0.39, *p*= 0.004; *r_s_* = 0.33, *p* = 0.031, respectively). In the HCTD-subgroup hEDS, disability was highly significantly positively correlated to fatigue reported by parent-proxy, pain and general health (*r_s_* = 0.74, *p* < 0.001; *r_s_* = 0.67, *p* < 0.001; *r_s_* = 0.65, *p* = 0.004, respectively).

## 4. Discussion 

This is the first quantitative, multicenter survey study to report on the burden of disease in childhood HCTD. Compared to normative data, children and adolescents with HCTD and HCTD-subgroups MFS, LDS, EDS and hEDS reported increased fatigue, pain, disability and decreased general health, with most differences translating into very large-sized effects. These results match our previous qualitative studies on children and adolescents with MFS [9,10] reporting on limitations in activities and participation, and descriptive studies in children with MFS, EDS and hEDS reporting on fatigue, pain and the negative impact on daily (physical) functioning [7,11,12,13,14,15,16,17,20]. Our results are also in line with studies using standardized validated measures in children with hEDS and HSD reporting on increased fatigue and pain [18], generalized hyperalgesia [21] and disability [17].

To interpret whether our data’s large effect sizes are clinically relevant, we referenced our data to effect size/minimal clinically important difference (MCID) benchmarks and the reported MCIDs of the used questionnaires in this study. Although MCID, defined as the smallest (absolute) difference in score that patients perceive as beneficial, is mainly used to evaluate interventions, it might also indicate clinical relevance. One of the distribution-based calculations of an MCID uses a cut-off point of Cohen’s *d =* 0.5 [38]. In our data, most differences translated into very large-sized effects, suggesting meaningful data. Moreover, the previously published MCID of the CHAQ-DI (median score 0.13 [35,39]) and PROMIS pediatric self-report (three points on the PROMIS T-score scale [40]) suggest that fatigue and disability in our study are clinically relevant in the HCTD-group and HCTD-subgroups EDS and hEDS. The most adverse sequels were reported in children with hEDS, whereas the least were reported in those with MFS. In previous qualitative studies, children and adolescents with MFS showed the ability to use productive coping strategies [9,10]. Articles on coping strategies in children with hEDS are lacking. A difference in coping strategies between the HCTD-subgroups might partly explain the results. Furthermore, participants with MFS, LDS and EDS were recruited by one of the Expert Centers for Marfan syndrome and related Connective Tissue Disorders in the Netherlands and Belgium. These participants receive medical care from a multidisciplinary team. Participants with hEDS were informed by the MFS and EDS patient associations and contacted the research team themselves if they were interested in participating in the study. Problems in the daily life of children with hEDS might therefore not have been discussed or treated in a clinical setting. This might contribute to further deterioration of physical functioning [20]. This new knowledge calls for systematic monitoring and standardized questionnaire assessments of fatigue, pain, disability and general health in the HCTD-group and HCTD-subgroups. Physical therapy, psychological counselling [18] and a multidisciplinary rehabilitation program [17] were reported as helpful in children and adolescents with hEDS and HSD [18]. This also indicates the importance of standardized physical assessments and tailored physical interventions in clinical care according to the Frequency, Intensity, Type and Time (FITT) factors [41] combined with lifestyle/sports education and psychosocial support.

A strength of our study is that a large sample of children with HCTD was included in our study. Furthermore, participants were recruited from one of the Expert Centers for Marfan syndrome and related Connective Tissue Disorders in the Netherlands, including the University Medical Centers of Amsterdam, Leiden, Groningen, Maastricht and Nijmegen, and at the Center for Medical Genetics of the Ghent University Hospital in Belgium. These countries have similar cultures and the surveys in Dutch and Belgium–Flemish are comparable. 

Our results must also be viewed within the limitations of the study. First, the sample sizes of the HCTD-subgroups EDS and LDS were small, and the explorative results should be interpreted with caution. In future, to demonstrate more subtle differences among HCTD-subgroups, a large European study with the cooperation of the European Reference Network (ERN) Skin, Mendelian Connective Tissue Disorders and ERN VASCERN could increase the sample size. Second, the children’s medical diagnosis was parent-reported. Our study design did not allow for checking the children’s medical diagnosis or clinical features. In 2017, the international clinical criteria for hEDS were revised, allowing for a better distinction from other joint hypermobility disorders [3]. Although our data were gathered in 2020, it is plausible that children with hEDS were not re-diagnosed and some might consequently not meet the 2017 hEDS criteria.

## 5. Conclusions

Children and adolescents with HCTD reported increased fatigue, pain, disability and decreased general health compared to normative data, with most differences translating to clinically relevant and very large-sized effects. The most adverse sequels were reported in children with hEDS, whereas the least were reported in those with MFS. 

This new knowledge calls for systematic monitoring and standardized assessments of fatigue, pain, disability and general health, not only through questionnaires but also physical assessments, and tailored interventions in clinical care. 

## Figures and Tables

**Figure 1 genes-12-00831-f001:**
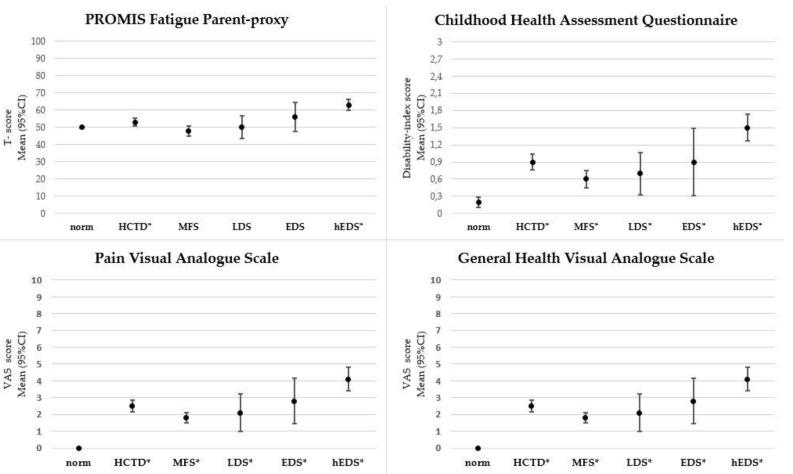
Legend: This figure shows the mean 95% confidence interval (CI) of the Childhood Health Assessment Questionnaire disability index (CHAQ-DI) score (range 0–3), PROMIS Fatigue Parent Proxy T-score (range 0–100), pain Visual Analogue Scale (VAS) score (range 0–10) and general health VAS score (range 0–10) of the normative data, the Heritable Connective Tissue Disorders (HCTD) group, and the following HCTD-subgroups: Marfan syndrome (MFS), Loeys-Dietz syndrome (LDS), Ehlers-Danlos Syndromes (EDS) and hypermobile Ehlers-Danlos syndrome (hEDS). * Compared to the normative data, this HCTD (sub)-group shows a significant difference.

**Table 1 genes-12-00831-t001:** Participants: sex, age and nationality of children of the HCTD-group, the HCTD-subgroups: MFS, LDS, EDS, hEDS; and of the parents who completed the survey.

Child	HCTD	MFS	LDS	EDS	hEDS	Among HCTD- Subgroups *p*-Value
n (%)	107 (100)	62 (58)	7 (7)	9 (8)	29 (27)	
Sex, n (%), Female	48 (45)	20 (32)	5 (71)	6 (67)	17 (59)	0.02 *
Age in years, M (SD)	10.2 (4.0)	10.1 (4.1)	11.0 (3.9)	10.8 (4.8)	10.0 (3.7)	0.90
Nationality, n (%)						0.03 *
Dutch	97 (90)	56 (90)	6 (86)	6 (67)	29 (100)	
Belgium	10 (10)	6 (10)	1 (14)	3 (33)	0 (0)	
Parent						
n (%)	107 (100)	62 (58)	7 (7)	9 (8)	29 (27)	
Sex, n (%), Female	86 (80)	46 (74)	6 (86)	7 (78)	27 (93)	0.20
Age in years, M (SD) ^a^	42,5 (7.1)	43.5 (8.1)	42.3 (6.3)	42.6 (4.2)	40.2 (4.9)	0.18
Nationality, n (%)						0.03 *
Dutch	97 (90)	56 (90)	6 (86)	6 (67)	29 (100)	
Belgium	10 (10)	6 (10)	1 (14)	3 (33)	0 (0)	

EDS, Ehlers-Danlos syndromes; HCTD, Heritable Connective Tissue Disorders; hEDS, hypermobile Ehlers-Danlos syndrome; LDS, Loeys-Dietz syndrome; M, mean; MFS, Marfan syndrome; n, number; SD, standard deviation; ^a^ missing 2; * significant difference.

**Table 2 genes-12-00831-t002:** The Patient Reported Outcomes Measurement Information System (PROMIS) Fatigue 10a–Pediatric v2.0 Short form and the PROMIS Fatigue 10a–Parent Proxy v2.0 Short form: T-scores of normative data, the HCTD-group and the HCTD-subgroups: MFS, LDS, EDS and hEDS; comparisons between normative T-scores and HCTD-group T-scores; and comparisons among HCTD-subgroup T-scores.

PROMIS Fatigue Pediatric Self-Report 8–18 Years	HCTD	norm	HCTD vs Norm *p*-Value	Effect size Cohen’s d/OR(95%CI)	MFS	LDS	EDS	hEDS	Among HCTD-Subgroups *p*-Value
n (%)	62 (100) ^a^	3042 (100)			36 (58) ^b^	5 (8)	6 (10) ^c^	15 (24) ^c^	
Sex, n (%), Female	30 (48)	1578 (52)	0.6		14 (39)	4 (80)	4 (67)	8 (53)	0.24
Age-groups, n (%)			0.08 *						0.50
8–12 years	26 (42)	1616 (53)							
13–18 years	36 (58)	1426 (47)							
T scores, M (SD)	49 (13)	50 (10)	0.44		44 (11)	47 (8)	52 (19)	61 (9)	<0.001 *
T score > 70, n (%)	8 (13)	213 (7)	0.07 *	2.0 (0.94–4.1)	1 (3)	0	2 (33)	5 (33)	0.02 *
PROMIS Fatigue Parent proxy 4–18 years									
n (%)	98 (100) ^a^	1980			57 (58) ^b^	7	9	25 (26) ^c^	
T scores, M (SD)	53 (12)	50 (10)	0.004 *	0.27	48 (11)	50 (9)	56 (13)	63 (8)	<0.001 *
T score > 70, n (%)	17 (16)	138 (7)	<0.001 *	2.83 (1.63–4.9)	3 (5)	0 (0)	3 (33)	11 (38)	0.023 *

CI, Confidence Interval; EDS, Ehlers-Danlos Syndromes; HCTD, Heritable Connective Tissue Disorders; hEDS, hypermobile Ehlers-Danlos syndrome; LDS, Loeys-Dietz Syndromes; MFS, Marfan syndrome; n, number; OR, Odds Ratio; p, probability; PROMIS, Patient Reported Outcomes Measurement Information System; SD, standard deviation; ^a^ missing 9; ^b^ missing 5; ^c^ missing 2; * significant difference.

**Table 3 genes-12-00831-t003:** Childhood Health Assessment Questionnaire (CHAQ), pain VAS and general health VAS: CHAQ domain scores, CHAQ disability-index scores, pain VAS scores and general health VAS scores; of normative data, the HCTD-group and the HCTD-subgroups: MFS, LDS, EDS and hEDS; comparisons between normative scores and HCTD-group scores; and among HCTD-subgroup scores.

CHAQ	HCTD	norm	HCTD vs Norm *p*-Value	Effect Size Cohen’s d	MFS	LDS	EDS	hEDS	Among HCTD-Subgroups *p*-Value
n (%)	99 (100) ^a^	80			58 (59) ^b^	7 (7)	9 (9)	25 (25) ^b^	
Sex, n (%) Female	44 (45)	33 (41)	0.59		19 (33)	5 (71)	6 (67)	14 (56)	0.04
Age in years, M (SD)	10.1 (4.1)	8.1 (3.6)	<0.001 *	0.49	10.1 (4.2)	11.0 (3.9)	10.8 (4.8)	9.6 (3.7)	0.81
Domain scores (0–3) M (SD)									
Dressing	0.8 (1.0)	0.5 (0.8)	0.03 *	0.33	0.6 (.9)	0.4 (0.8)	1.0 (0.9)	1.3 (1.3)	
Arising	0.7 (.8)	0.1 (0.3)	<0.001 *	0.99	0.4 (.7)	0.9 (0.9)	0.7 (0.8)	1.4 (0.9)	
Eating	1.0 (.9)	0.4 (0.7)	<0.001 *	0.74	0.7 (.8)	0.7 (0.8)	0.9 (0.9)	1.7 (0.9)	
Walking	0.6 (.9)	0.0 (0.1)	<0.001 *	0.94	0.3 (.7)	0.1 (0.4)	0.6 (0.9)	1.3 (0.9)	
Hygiene	0.8 (1.0)	0.3 (0.6)	<0.001 *	0.60	0.5 (.8)	0.1 (0.4)	0.8 (0.9)	1.4 (1.1)	
Reach	0.9 (.8)	0.2 (0.5)	<0.001 *	1.0	0.5 (.8)	0.9 (0.7)	0.9 (0.9)	1.5 (0.6)	
Grip	1.1 (.9)	0.2 (0.6)	<0.001 *	1.18	0.8 (.8)	1.1 (0.9)	1.1 (1.1)	1.7 (0.9)	
Activity	1.1 (1.0)	0.2 (0.5)	<0.001 *	1.14	0.7 (.8)	1.1 (0.7)	1.4 (1.2)	1.9 (0.7)	
CHAQ-DI scores (0–3)									
M (SD); median (IQR)	0.9 (0.7); 0.6 (1.1)	0.2 (0.4)	<0.001 *	1.23	0.6 (0.6); 0.4 (0.8)	0.7 (0.5); 0.6 (0.6)	0.9 (0.9); 0.5 (1.8)	1.5 (0.6); 1.5 (1.1)	<0.001 *
Pain VAS scores (0–10)									
n (%)	93 (100) ^c^	80			54 (58) ^a^	7 (7)	9 (10)	23 (25) ^d^	
M (SD); median (IQR)	2.8 (3.1); 1.9 (5.5)	0.0 (0.2)	<0.001 *	1.27	1.3 (2.3); 0 (2.0)	2.8 (2.7); 2.5 (5.7)	3.6 (2.2); 2.9 (3.2)	6.1(2.7); 7.0 (3.7)	<0.001 *
General health VAS scores (0–10)									
n (%)	82 (100) ^e^	80			48 (59) ^f^	6 (7) ^g^	6 (7) ^h^	22 (27) ^i^	
M (SD); median (IQR)	2.5 (1.8); 2.0 (2.5)	0.0 (0.1)	<0.001 *	2.04	1.8 (1.2); 1.5 (1.6)	2.1 (1.5); 2.4 (2.5)	2.8 (2.1); 2.3 (1.7)	4.1 (1.8); 4.1 (3.3)	<0.001 *

CHAQ, Childhood Health Assessment Questionnaire; CHAQ DI, Disability-index; *d*, Cohen’s d effect size; EDS, Ehlers-Danlos Syndromes; HCTD, Heritable Connective Tissue Disorders; hEDS, hypermobile Ehlers-Danlos syndrome; IQR, interquartile range; LDS, Loeys-Dietz Syndromes; MFS, Marfan syndrome; n, number; p, probability; SD, standard deviation; VAS, Visual Analog Scale; ^a^ missing 8; ^b^ missing 4; ^c^ missing 14; ^d^ missing 6; ^e^ missing 25; ^f^ missing 14; ^g^ missing 1; ^h^ missing 3; ^i^ missing 7; * significant difference.

## Data Availability

All data are available on request from the corresponding author.

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
