# Peer review of "Heritable Connective Tissue Disorders in Childhood: Increased Fatigue, Pain, Disability and Decreased General Health"

_genes, 2021, doi:10.3390/genes12060831_

Round 1
Reviewer 1 Report
Warnink-Kavelaars et al did a good job describing the common disability features of children with connective tissue disorders. The manuscript is going to be a great resource for healthcare providers and parents.
I don't have any comments and recommending the manuscript to be accepted in its present form.
Author Response
Dear reviewer,
Thank you very much for you time, effort and comments.
The English language and style as well as the tables and figure of the manuscript are spell check and revised.
Kind regards,
Jessica Warnink-Kavelaars
Pediatric Rehabilitation Physician
Department of Rehabilitation MEdicine
Amsterdam UMC, Amsterdam, the Netherlands

Reviewer 2 Report
This is a well written, interesting manuscript, employing appropriate methodology that will contribute to the literature.
Author Response
Dear reviewer,
Thank you very much for you time, effort and comments.
The English language and style as well as the tables and figure of the manuscript are spell check and revised.
Kind regards,
Jessica Warnink-Kavelaars
Pediatric Rehabilitation Physician
Department of Rehabilitation Medicine
Amsterdam UMC, Amsterdam, the Netherlands

Reviewer 3 Report
The paper is OK
Author Response

(The authors gave the same response as above.)

Reviewer 4 Report
The paper is very interesting and well written. The manuscript presents the first, quantitative, multicenter survey study to report on the burden of 270 disease in childhood HCTD. It's well written and very clear. It could be very useful in order to stimulate further studies.
Author Response
Dear reviewer,
Thank you very much for you time, effort and comments.
The English language and style as well as the tables and figure of the manuscript are spell checked and revised.
The Follow You research team and the Pediatric Heritable Connective Tissue Disorders Study Group, will continue to develop physical assessments, and tailored interventions in clinical care for children and adolescents with HCTD.
Kind regards,
Jessica Warnink-Kavelaars
Pediatric Rehabilitation Physician
Department of Rehabilitation Medicine
Amsterdam UMC, Amsterdam, the Netherlands

Reviewer 5 Report
this interesting study reported the correlation of connective diseases with a novel panel of physical features
Major concern regards the fact that connective diseases have different Clinical features based on Genes causing disorders should be a great insight if the authors can also add this information to their analysis
Author Response
Dear reviewer,
Thank you very much for you time, effort and comments.
The English language and style as well as the tables and figure of the manuscript are spell checked and revised.
I share your concern. Heritable Connective Tissue Disorders (HCTD) are characterised by pathological connective tissue fragility and the diagnosis is, indeed, based on different clinical criteria and/or molecular confirmations by a causative genetic variant. The study design did not allow checking on the children’s medical diagnosis or clinical features. It would be very interesting to add this research question to future research.
The Follow You research team and the Pediatric Heritable Connective Tissue Disorders Study Group will continue to develop physical assessments and tailored interventions in clinical care for children and adolescents with HCTD.
Kind regards,
Jessica Warnink-Kavelaars
Pediatric Rehabilitation Physician
Department of Rehabilitation Medicine
Amsterdam UMC, Amsterdam, the Netherlands
